# Effects of nitrogen, phosphorus and potassium formula fertilization on the yield and berry quality of blueberry

Xinyu Zhang[1], Shuangshuang Li[2], Xiaoli An[1], Zejun Song[1], Yunzheng Zhu[1], Yi Tan[1], Xiaolan Guo[3], Delu Wang®[1] *

1 College of Forestry, Guizhou University, Huaxi, Guiyang, Guizhou, China, 2 General Office of Guizhou Provincial Party Committee, Guiyang, Guizhou, China, 3 College of Life Sciences, Huizhou University, Huicheng, Huizhou, Guangdong, China

* deluwang23@aliyun.com

**Data Availability Statement:** All relevant data are within the paper.

**Funding:** Author thanks the National Natural Science Foundation of China (NSFC 31260192) for

## Abstract

Through the application ratio of nitrogen (N), phosphorus (P) and potassium (K) in the field, $L_9$ ($3^3$) orthogonal experimental design was used to study the effects of different N, P and K ratios on the yield and quality of blueberry fruit, aiming to optimize the amount of supplied fertilizers. The results showed that N, P and K fertilizer had different effects on fruit yield and quality, among which K fertilizer was the most important factor. Fertilization could significantly improve the yield and fruit quality of blueberry, and the average yield of fertilization treatment was 37.78% higher than that of the control group (CK). Even the treatment with the worst results F6 (N2P3K1), its single fruit weight, anthocyanins, total phenols, soluble solids and soluble protein content were 1.09, 1.32, 1.23, 1.08 and 1.21 times higher than the control (CK), respectively. Based on the comprehensive evaluation of principal component analysis and multi factor analysis of variance, the best fertilization combination for high-yield and good-quality blueberries was N1P2K2 (F2), that is, the best fertilization effect was that including N 100 g/plant, $P_2O_5$ 25 g/plant, $K_2O$ 25 g/plant, applied in the form of ammonium sulfate (472 g/plant), superphosphate (41 g/plant) and potassium sulfate (40 g/plant), respectively.

## Introduction

Nitrogen (N), phosphorus (P) and potassium (K) fertilizers are well-known mineral elements necessary for plant growth and development [1], and the application of fertilizers containing these elements can significantly improve the yield and quality of fruit trees [2]. However, in actual production, the proportion of N, P and K is often unbalanced due to the lack of scientific ratio, which affects the absorption and utilization of nutrients by plants, reduces the yield and quality, and increases the risk of nutrient loss and environmental pollution [3]. Reasonable ratio of N, P and K can significantly promote plant growth while relatively reducing the amount of various fertilizers [4]. In the application of proportion-based fertilization, the optimal ratio of N, P and K and the appropriate amount of application should be determined by

their financial support. The funders had no role in study design, data collection and analysis, decision to publish, or preparation of the manuscript.

**Competing interests:** The authors have declared that no competing interests exist.

combining the fertilizer requirement rule of species, fertilizer supply characteristic of growing soil, fertilizer effect and nutrient interaction rule [5]. However, due to the great differences in climate, soil fertility and texture, and crop varieties in China, the amount and proportion of N, P and K fertilizers applied have been greatly affected, and the situation of improper fertilizer use still exists [3]. Moreover, no fertilization model has been found to be suitable for all plants, fertilizers and regions [6].

The cultivated area and yield of blueberries in China have jumped to the first place in the world. Guizhou Province is one of the main producing areas, with an area of 15,000 hm$^2$ and a yield of 85,000 tons, but the yield per unit area and quality are not high [7]. This is closely related to the cultivation technology of blueberry, especially to the substandard fertilization management technology of blueberry [8]. However, a scientific and reasonable fertilization method can effectively promote the growth of blueberry plants [9] and obtain blueberries with high quality and yield [10]. As blueberry is an oligotrophic species with shallow root system [11, 12], it prefers ammonium N [13] and has low requirements for P, K, Ca and Mg [14], so there is often a phenomenon of overapplying N fertilizer and underapplying P and K fertilizer. At the same time, due to the poor absorption capacity of its roots, it is sensitive to fertilization [15]. Insufficient or excessive fertilization affect negatively the yield and quality of blueberries [16], and the vegetative growth of blueberries is inhibited, or even damaged [17]. Previous studies have shown that fertilization has a significant effect on the yield and quality of blueberries. For example, applying wood compost to blueberries cannot only promote their growth and increase their yield [18], but also it can improve their fruit soluble solid content [19]. Albert et al. [10]pointed out that the application of N, P and K fertilizers could increase the anthocyanin content of blueberry fruits. Reasonable combination of N, P and K application can increase blueberry yield [14]. However, some studies have pointed out that applying conventional fertilizers without guidance, can lead to salt accumulation in soil and reduce the yield and fruit quality of blueberries [8]. Studies on combined application of N, P and K have been fruitful in crops and fruit trees [3, 20, 21]. However, researches on blueberry mainly focus on the effect of single-nutrient fertilizer, while there are few studies on the interaction between N, P and K and how they affect the yield and quality of blueberry.

Therefore, this paper takes 5-year-old rabbiteye blueberry cv. Brilliant plants, explores the ratio of N, P and K applied to blueberry in this region through orthogonal test, aiming to optimize the appropriate amount of blueberry fertilizer by using comprehensive evaluation and variance analysis, in order to provide technical support for economic fertilizer saving, yield and quality improvement, and standardized cultivation of blueberry.

## Materials and methods

### Experimental site

The experimental site is located in Wuyang hemp blueberry plantation, Xuanwei Town, Majjiang County, Guizhou Province, China, between 26°21′-26°31′N and 107°33′-107°47′E, which is characterized by subtropical monsoon humid climate. The annual average temperature is 15.7°C, the annual average precipitation is 1266 mm, and the annual average frost-free period is 293 days. At an average altitude of 670 m. The base soil was an acidic yellow type with pH 4.35–5.50, organic matter content 23.90 g•kg$^{-1}$, total nitrogen (N) content 7.84%, total phosphorus (P) content 1.9%, and total potassium (K) content 28%.

### Plant material

Five -year-old rabbiteye blueberry (*Vacciniumvirgatum* cv. Brightwell) plants with similar growth, were selected as the experimental material.

## Experimental design

The experiment was carried out under field conditions. Fertilizers containing N, P and K doses at three doses for each nutrient. Each factor was set at three levels according to Guo et al. [22], and each treatment was set with three replicates, three plants for each replicate. the plantation distancesplant spacing was is 1.5m, and the plantation density is 4444 trees/ha. Non fertilized plants were used as control (CK), and the experiment adopted $L_9$ ($3^3$) orthogonal design. The fertilizers used were ammonium sulfate (N 21.2%), superphosphate ($P_2O_5$ 60.6%) and potassium sulfate ($K_2O$ 63.2%), all of high chemical purity. The field was averagely fertilized four times throughout the year: Early March (pre-bloom), Early May (before fruit production), Late August to Early September (after fruit production, flower bud differentiation stage) and Early December (reductive fertilizer). The fertilizers were applied into a trench 50 cm in length, 20 cm in width and 20 cm in depth on each side of the periphery of the tree canopy projection, mixed and covered with soil. The experiment started in December 2014, and the experimental design is shown in Tables 1 and 2.

## Method of index determination

**The sample collection.**　At the peak of ripening period, the fruits were collected from the periphery of trees, with three replicates per treatment and three plants per replicate, and 60 fruits were collected from each plant, which were brought back to the laboratory under ice bath condition in reserve.

**Determination of fruit indexes.**　Single fruit weight was measured by using a 1/10000 scale; Yield per plant = number of blueberry fruits per plant * average weight per fruit. The anthocyanin content was determined according to the method of Barnes et al. [23]. The Folin-Ciocalteu reagent method was used to establish a standard curve with gallic acid as the

**Table 1. Test factor level (annual application rate).**

| factor | factor A | factor B | factor C |
|---|---|---|---|
| level | N(g/plant) | P(g/plant) | K(g/plant) |
| 1 | 100(472) | 50(83) | 100(158) |
| 2(1/2) | 50(236) | 25(41) | 50(79) |
| 3(1/4) | 25(118) | 12.5(21) | 25(40) |

Table notes Data in parentheses are compound dosage g/plant.

**Table 2. Treatments of N, P and K with their dosages on blueberry plants.**

| Code | Treatment | N(g/plant) | $P_2O_5$(g/plant) | $K_2O$(g/plant) |
|---|---|---|---|---|
| F1 | N1P1K1 | 100 | 50 | 100 |
| F2 | N1P2K2 | 100 | 25 | 50 |
| F3 | N1P3K3 | 100 | 12.5 | 25 |
| F4 | N2P1K2 | 50 | 50 | 50 |
| F5 | N2P2K3 | 50 | 25 | 25 |
| F6 | N2P3K1 | 50 | 12.5 | 100 |
| F7 | N3P1K3 | 25 | 50 | 25 |
| F8 | N3P2K1 | 25 | 25 | 100 |
| F9 | N3P3K2 | 25 | 12.5 | 50 |
| CK | N0P0K0 | 0 | 0 | 0 |

standard substance, and the content was expressed as (mg/g FW) [24]. The flavonoid content was appropriately determined according to the method of Wolfe et al. [25], the standard curve was established with catechin as the standard substance, and the content was expressed as (mg/g FW). The soluble sugar content was determined by HPLC system, the titratable acid content was detected by ultra-fast liquid chromatography combined with photodiode array [26], and the soluble solid content was determined by handheld refractometer(SW20) [27]. The sugar-acid ratio is the ratio of soluble solids to titratable acids.

## The data processing

Excel 2013 and SPSS 18 software programs were used for statistical analysis of the experimental data, ANOVA and least significant difference (LSD) were applied to compare the differences between different data groups, the K value and range R of different factors at different levels was calculated and the principal component analysis method for comprehensive evaluation was used.

## Results

### Effects of different formulations of fertilization on blueberry fruit yield

There were significant differences in the yield of blueberry fruits among different fertilization treatments. The weight of single blueberry fruit in fertilization treatment was higher than CK, and the most obvious effects were observed in F2 and F6 treatments, which gave 8.67% higher values than CK. The yield of blueberry per plant of fertilization treatment was significantly higher than that of CK, which was F2, F4, F8 and F9. The most effective was F2 treatment, which was 79.26% higher than CK. Range analysis showed that N2P3K2 and N3P3K2 were the best fertilizer combinations affecting the fruit weight and yield per plant of blueberry. According to the R value, the order of influence of the three factors on the single fruit weight and yield per plant of blueberry is P>N>K and P>K>N, respectively (Fig 1, Table 3).

### Effect of different formula fertilization on blueberry fruit quality

Formula fertilization caused significant changes in the quality of blueberry fruit, with different effects. Among the eight main quality indicators, anthocyanin, total phenol, soluble solids, soluble sugar content and sugar: acid ratio increased overall, titratable acid content decreased overall, flavonoid and soluble protein content some increased and some decreased (Fig 2).

Compared with CK, the anthocyanin content, total phenol content and soluble solid content of blueberry fruits treated with fertilizer increased by 9.5% ˜ 62.96%, 14.94% ˜ 31.03% and 5.89% ˜ 23.65% respectively (Fig 2A, 2B and 2E).

The effect of Fertilization on the sugar: acid ratio of blueberry fruit was similar to that of soluble sugar content. Except for treatment F6, the soluble sugar content and sugar: acid ratio in blueberry fruits of other fertilization treatments were higher than CK. The most obvious effect was treatment F2, which was significantly different from CK, 57.19% and 48.64% higher than CK respectively (Fig 2F and 2H). The effect of Fertilization on the titratable acid content of blueberry fruit is opposite to that of soluble sugar. Except for treatment F6, the titratable acidity of other fertilization treatments is lower than CK, but there is no significant difference with CK. The most obvious effect was that of treatment F9(17.07% lower than CK) (Fig 2G).

Except for treatments F5, F6 and F8, the flavonoid content of other fertilization treatments was higher than CK, of which treatment F2 was significantly higher (36.36%) than CK (Fig 2C). The effect of Fertilization on the soluble protein content of blueberry fruit is different from that of flavonoids. Except for treatments F1, F2, F3 and F4, the soluble protein content of

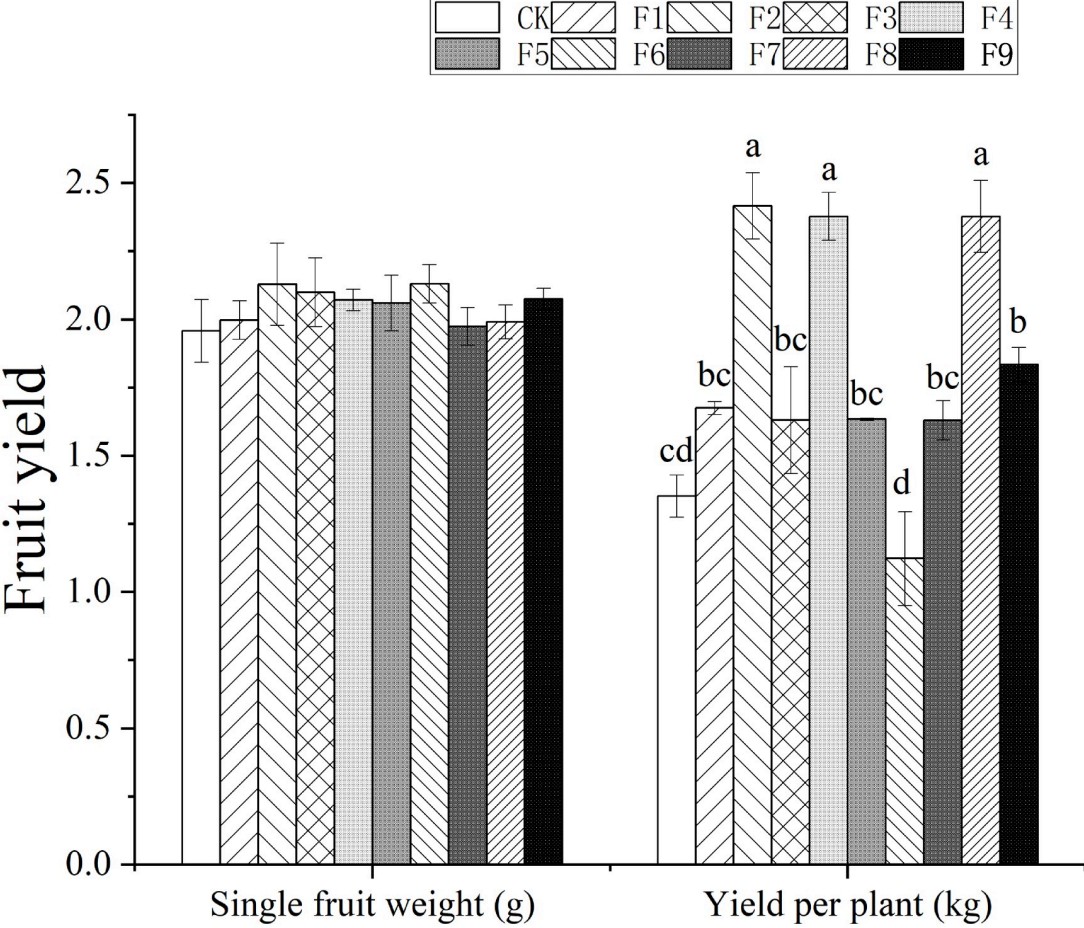

**Fig 1. Effects of different formulations of fertilization on blueberry yield.** Figure notes Different lowercase letters in the figure indicate significant differences between treatments(P<0.05).

other fertilization treatments is higher than CK, of which treatment F6 is significantly higher (20.59%) than CK (Fig 2D).

The range analysis showed that the optimal combination affecting the content of anthocyanins, total phenols, flavonoids, soluble protein, soluble solids, soluble sugar, titratable acidity and sugar acid ratio of blueberry fruits were N2P1K3, N3P3K1, N1P1K2, N3P3K1, N1P1K2 or N2P1K2, N3P2K2, N1P2K2 and N1P2K2, respectively. According to the R value, it can be judged that the order of influence of the three factors on the above-mentioned parameters of blueberry fruit is as follows: P>K>N,N>K>P, K>N>P,N>K>P,N>K = P,K>N>P,K>N>P and K>N>P, respectively. According to the F value, the three factors of N, P and K fertilizer had a very significant effect on the total phenol of blueberry fruit (P<0.01), and a significant effect on the soluble sugar content of blueberry fruit (P<0.05) (Table 4).

## Comprehensive analysis of the effect of formula fertilization on the yield and berry quality of blueberry

The yield and berry quality of blueberry were evaluated by principal component analysis. The results show that according to the principle that the eigenvalue is greater than 1, the

**Table 3. Results and analysis of orthogonal experiment on fruit yield of blueberry.**

|  | Code | N(A) | P(B) | K(C) | Single fruit weight (g) | Yield per plant (kg) |
|---|---|---|---|---|---|---|
|  | F1 | 1 | 1 | 1 | 2.00 | 1.68 |
|  | F2 | 1 | 2 | 2 | 2.13 | 2.42 |
|  | F3 | 1 | 3 | 3 | 2.10 | 1.63 |
|  | F4 | 2 | 1 | 2 | 2.07 | 2.38 |
|  | F5 | 2 | 2 | 3 | 2.06 | 1.63 |
|  | F6 | 2 | 3 | 1 | 2.13 | 1.12 |
|  | F7 | 3 | 1 | 3 | 1.97 | 1.63 |
|  | F8 | 3 | 2 | 1 | 1.99 | 2.38 |
|  | F9 | 3 | 3 | 2 | 2.07 | 1.83 |
|  | CK | 0 | 0 | 0 | 1.96 | 1.35 |
| *Singlefruitweight(g)* | K1 | 2.08 | 2.01 | 2.04 | P>N>K | |
|  | K2 | 2.09 | 2.06 | 2.09 | | |
|  | K3 | 2.01 | 2.10 | 2.04 | | |
|  | R | 0.08 | 0.09 | 0.05 | | |
|  | F value | 5.024 | 6.005 | 2.618 | | |
| *Yieldperplant(kg)* | K1 | 1.91 | 1.89 | 1.73 | P>K>N | |
|  | K2 | 1.71 | 2.14 | 2.21 | | |
|  | K3 | 1.95 | 1.53 | 1.63 | | |
|  | R | 0.24 | 0.61 | 0.58 | | |
|  | F value | 0.28 | 1.70 | 1.72 | | |

cumulative contribution rate of the first three selected component factors reaches 78.89%. Among the three principal components, the characteristic value of the first principal component is 4.81, and the variance contribution rate is 48.14%, which mainly reflects the single plant yield, flavonoids, soluble protein, soluble solids, soluble sugar, titratable acid content and sugar: acid ratio. The second and third principal components mainly reflect the single fruit weight, anthocyanin and total phenol content (Table 5).

Taking the variance of each principal component as the weight, the comprehensive scores of the three principal components of blueberry fruit quality and yield indicators under 10 treatments were calculated and ranked. It was concluded that the effect of fertilization treatment F2 was better than other treatments, the effect of F4 treatment was ranked second (Table 6), and the data results of the second year also proved this conclusion (Table 7). In addition, the benefit-cost ratio of treating F2 and F4 is 16.13 and 15.87, respectively, so F2 treatment is the best.

From the comprehensive score Y value, it can be seen that fertilization treatment is higher than CK treatment, indicating that formula fertilization can effectively promote the yield and berry quality of blueberry. Range analysis showed that the best combination affecting the yield and berry quality of blueberry was N1P2K2 treatment. According to the R value, the order of influence of the three factors on blueberry yield and berry quality is K>P>N (Table 6).

## Discussion

The reasonable combination of N, P and K can effectively improve the yield and fruit quality of trees. Studies have shown that these nutrients and their interactions have a remarkable influence on citrus yield, quality and soil residual available nutrients [3]; The combined application of N, P and K can increase the yield and improve fruit quality of kiwifruit [28]. The

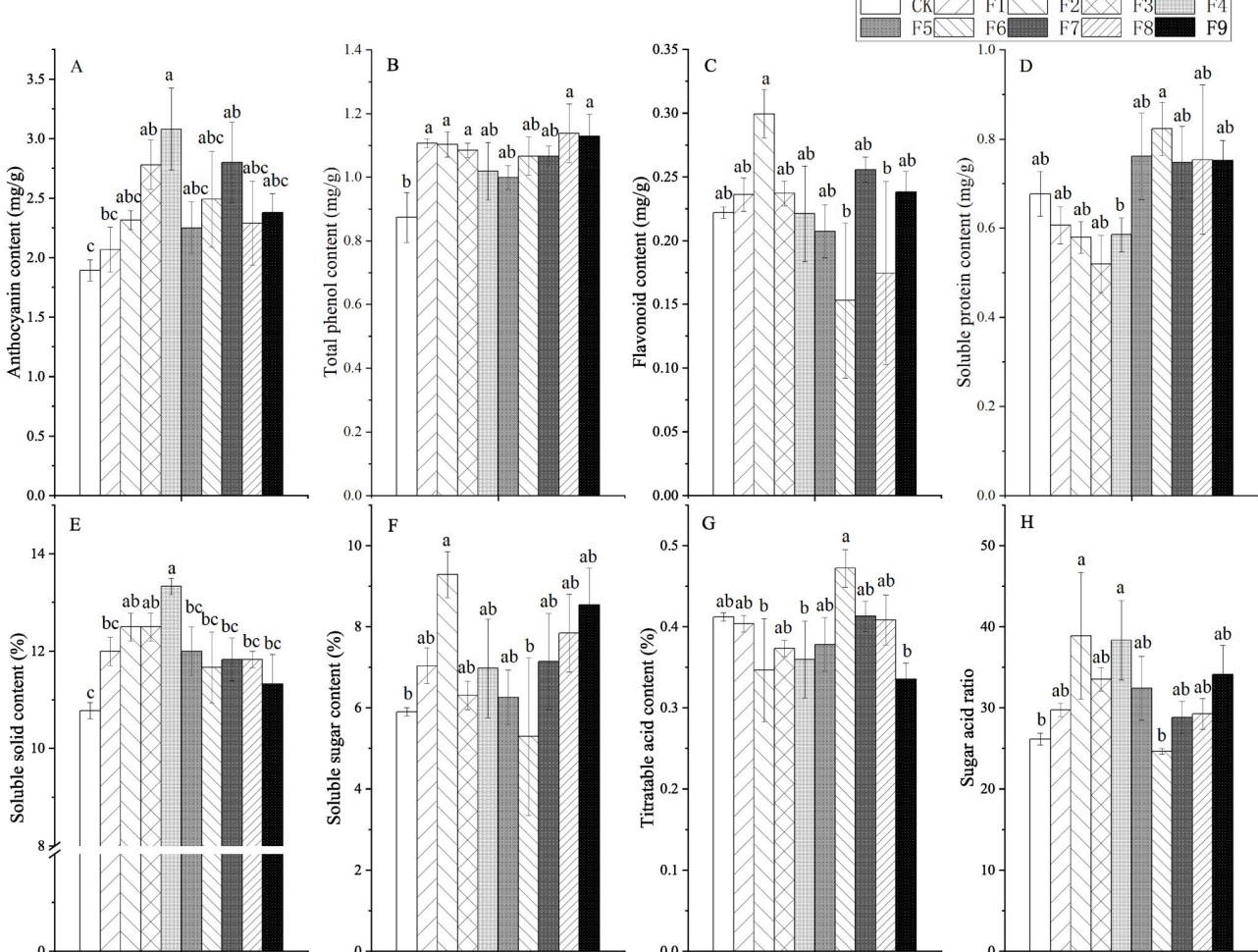

**Fig 2. Effect of different fertilization formulas on blueberry fruit quality.** Figure notes Different lowercase letters in the figure indicate significant differences between treatments(P<0.05).

reasonable application of N, P and K fertilizers can ensure the nutritional balance of blueberries and improve the fruit yield [22]. The results of this experiment indicate that the combined applications of different N, P and K levels have a significant impact on the yield and quality of blueberry. From the comprehensive perspective of all indicators, the impact of K Fertilizer on the yield and quality of blueberry fruit is greater than that of N and P fertilizer, which may be due to the lack of K in the soil of this region.

K fertilizer is the most important factor affecting fruit quality [29]. In accordance, we found that K fertilizer had the most significant effect on the increment of soluble sugar and the decrement of titratable acidity in blueberry fruit. There was also significant improvement of the flavonoid content and sugar: acid ratio. It is consistent with the results of Wu Zhengchao et al. [30] who studied the effect of K fertilizer on the soluble sugar, Vc and sugar: acid ratio. Since K can participate in the metabolism and transportation of sugar, it increases the sugar content of fruit [31], and also promotes the transformation of phenolic substances into flavonoids by activating metabolic enzymes. N fertilizer was the most important factor affecting the content of total phenol, soluble protein and soluble solid in blueberry, and the effect of factor N on the

**Table 4. Orthogonal test on results and analysis of orthogonal test on blueberry fruit quality.**

| | Code | N(A) | P(B) | K(C) | Anthocyanin (mg/g) | Total phenol (mg/g) | Flavonoid (mg/g) | Soluble protein (mg/g) | Soluble solid (%) | Soluble sugar (%) | Titratable acidity (%) | Sugar acid ratio |
|---|---|---|---|---|---|---|---|---|---|---|---|---|
| | F1 | 1 | 1 | 1 | 2.07 | 1.11 | 0.236 | 0.61 | 12.00 | 7.04 | 0.40 | 29.75 |
| | F2 | 1 | 2 | 2 | 2.32 | 1.10 | 0.300 | 0.58 | 12.50 | 9.29 | 0.35 | 38.90 |
| | F3 | 1 | 3 | 3 | 2.78 | 1.08 | 0.237 | 0.52 | 12.50 | 6.31 | 0.37 | 33.55 |
| | F4 | 2 | 1 | 2 | 3.08 | 1.02 | 0.221 | 0.58 | 13.33 | 6.98 | 0.36 | 38.36 |
| | F5 | 2 | 2 | 3 | 2.25 | 1.00 | 0.207 | 0.76 | 12.00 | 6.26 | 0.38 | 32.41 |
| | F6 | 2 | 3 | 1 | 2.49 | 1.07 | 0.153 | 0.82 | 11.67 | 5.30 | 0.47 | 24.68 |
| | F7 | 3 | 1 | 3 | 2.80 | 1.07 | 0.256 | 0.75 | 11.83 | 7.15 | 0.41 | 28.82 |
| | F8 | 3 | 2 | 1 | 2.29 | 1.14 | 0.175 | 0.75 | 11.83 | 7.84 | 0.41 | 29.27 |
| | F9 | 3 | 3 | 2 | 2.38 | 1.13 | 0.238 | 0.75 | 11.33 | 8.54 | 0.34 | 34.15 |
| | CK | 0 | 0 | 0 | 1.89 | 0.87 | 0.222 | 0.68 | 10.78 | 5.91 | 0.41 | 26.17 |
| Anthocyanin (mg/g) | K1 | 2.39 | 2.65 | 2.28 | P>K>N | | | | | | | |
| | K2 | 2.61 | 2.29 | 2.59 | | | | | | | | |
| | K3 | 2.49 | 2.55 | 2.61 | | | | | | | | |
| | R | 0.22 | 0.36 | 0.33 | | | | | | | | |
| | F value | 0.199 | 0.592 | 0.565 | | | | | | | | |
| Totalphenol(mg/g) | K1 | 1.10 | 1.06 | 1.10 | N>K>P | | | | | | | |
| | K2 | 1.03 | 1.08 | 1.08 | | | | | | | | |
| | K3 | 1.11 | 1.09 | 1.05 | | | | | | | | |
| | R | 0.08 | 0.03 | 0.05 | | | | | | | | |
| | F value | 1506.599** | 166.362** | 554.471** | | | | | | | | |
| Flavonoid(mg/g) | K1 | 0.258 | 0.238 | 0.188 | K>N>P | | | | | | | |
| | K2 | 0.194 | 0.227 | 0.253 | | | | | | | | |
| | K3 | 0.223 | 0.210 | 0.233 | | | | | | | | |
| | R | 0.064 | 0.028 | 0.065 | | | | | | | | |
| | F value | 6.205 | 1.239 | 6.768 | | | | | | | | |
| Solubleprotein (mg/g) | K1 | 0.57 | 0.65 | 0.738 | N>K>P | | | | | | | |
| | K2 | 0.72 | 0.70 | 0.64 | | | | | | | | |
| | K3 | 0.75 | 0.70 | 0.68 | | | | | | | | |
| | R | 0.18 | 0.05 | 0.098 | | | | | | | | |
| | F value | 3.415 | 0.315 | 0.703 | | | | | | | | |
| Solublesolid(%) | K1 | 12.33 | 12.39 | 11.83 | N>K = P | | | | | | | |
| | K2 | 12.33 | 12.11 | 12.39 | | | | | | | | |
| | K3 | 11.67 | 11.83 | 12.11 | | | | | | | | |
| | R | 0.66 | 0.56 | 0.56 | | | | | | | | |
| | F value | 0.923 | 0.481 | 0.481 | | | | | | | | |
| Solublesugar(%) | K1 | 7.55 | 7.06 | 6.73 | K>N>P | | | | | | | |
| | K2 | 6.18 | 7.80 | 8.27 | | | | | | | | |
| | K3 | 7.84 | 6.72 | 6.57 | | | | | | | | |
| | R | 1.66 | 1.08 | 1.70 | | | | | | | | |
| | F value | 48.820* | 18.844* | 54.443* | | | | | | | | |

*(Continued)*

**Table 4.** (Continued)

| | Code | N(A) | P(B) | K(C) | Anthocyanin (mg/g) | Total phenol (mg/g) | Flavonoid (mg/g) | Soluble protein (mg/g) | Soluble solid (%) | Soluble sugar (%) | Titratable acidity (%) | Sugar acid ratio |
|---|---|---|---|---|---|---|---|---|---|---|---|---|
| *Titratableacidity (%)* | K1 | 0.37 | 0.39 | 0.43 | | | | K>N>P | | | | |
| | K2 | 0.40 | 0.38 | 0.35 | | | | | | | | |
| | K3 | 0.39 | 0.39 | 0.39 | | | | | | | | |
| | R | 0.03 | 0.01 | 0.08 | | | | | | | | |
| | F value | 0.526 | 0.19 | 4.042 | | | | | | | | |
| *Sugaracidratio* | K1 | 34.07 | 32.31 | 27.90 | | | | K>N>P | | | | |
| | K2 | 31.82 | 33.53 | 37.14 | | | | | | | | |
| | K3 | 30.75 | 30.79 | 31.59 | | | | | | | | |
| | R | 3.32 | 2.73 | 9.24 | | | | | | | | |
| | F value | 1.336 | 0.872 | 10.054 | | | | | | | | |

* indicates that the influence of factors reaches a significant level (P<0.05),

** Indicating that the influence of factors reached a very significant level (P<0.01).

content of total phenol in blueberry fruit was highly significant (P<0.01). It was shown that the application of N fertilizer had a significant impact on the fruit quality of blueberry. It was shown that the P fertilizer has a relatively obvious effect on the formation of blueberry anthocyanins, which may be because P mediates the accumulation of anthocyanins in plants by participating in the expression of anthocyanin synthesis genes [32]. P is involved in the metabolism of plants [33], and applying more (but not excessive) P fertilizer can increase the yield of blueberry [34]. In this study, P fertilizer is the biggest factor affecting blueberry yield, but it has no significant impact, which is consistent with the results of Laford et al. [35], because blueberry has very low demands for P, and the P in the soil can meet its demand. When P content in soil is high, increasing P fertilizer will reduce blueberry yield and delay fruit maturity [36]. Therefore, in order to improve the yield and berry quality of blueberry, it is

**Table 5. Eigenvector load matrix of three principal components.**

| Yield or quality index | Feature vector | Feature vector | Feature vector |
|---|---|---|---|
| | The 1st principal component | The 2nd principal component | The 3rd principal component |
| *Singlefruitweight* | 0.412 | 0.548 | 0.186 |
| *Yieldperplant* | 0.809 | -0.134 | 0.218 |
| *Anthocyanin* | 0.442 | 0.724 | -0.050 |
| *Totalphenol* | 0.442 | 0.109 | 0.774 |
| *Flavonoid* | 0.673 | -0.460 | -0.307 |
| *Solubleprotein* | -0.649 | -0.007 | 0.583 |
| *Solublesolid* | 0.738 | 0.571 | -0.140 |
| *Solublesugar* | 0.742 | -0.498 | 0.420 |
| *Titratableacidity* | 0.838 | -0.310 | -0.109 |
| *Sugaracidratio* | 0.963 | 0.028 | -0.108 |
| *Eigenvalue* | 4.814 | 1.737 | 1.337 |
| *Variancecontributionrate* | 48.141% | 17.374% | 13.374% |
| *Accumulativecontributionrate* | 48.141% | 65.515% | 78.889% |

**Table 6. Comprehensive score and ranking of yield and quality indexes of blueberry fruit under different fertilization treatments (2015).**

|  | Code | N(A) | P(B) | K(C) | Score of the 1st principal component factor | Score of the 2nd principal component factor | Score of the 3rd principal component factor | Comprehensive score Y | Comprehensive score ranking |
|---|---|---|---|---|---|---|---|---|---|
|  | F1 | 1 | 1 | 1 | 46.66 | 6.35 | -0.26 | 52.75 | 7 |
|  | F2 | 1 | 2 | 2 | 58.35 | 5.86 | -0.25 | 63.96 | 1 |
|  | F3 | 1 | 3 | 3 | 50.62 | 7.66 | -1.15 | 57.13 | 4 |
|  | F4 | 2 | 1 | 2 | 56.96 | 8.04 | -1.37 | 63.63 | 2 |
|  | F5 | 2 | 2 | 3 | 48.62 | 6.97 | -0.87 | 54.72 | 5 |
|  | F6 | 2 | 3 | 1 | 39.86 | 7.41 | -0.38 | 46.90 | 9 |
|  | F7 | 3 | 1 | 3 | 45.80 | 6.69 | -0.09 | 52.40 | 8 |
|  | F8 | 3 | 2 | 1 | 47.20 | 5.94 | 0.43 | 53.56 | 6 |
|  | F9 | 3 | 3 | 2 | 51.73 | 5.55 | 0.11 | 57.40 | 3 |
|  | CK | 0 | 0 | 0 | 41.07 | 6.00 | -0.38 | 46.69 | 10 |
| Y | K1 | 57.95 | 56.26 | 51.07 | K>P>N | | | | |
|  | K2 | 55.08 | 57.41 | 61.66 | | | | | |
|  | K3 | 54.45 | 53.81 | 54.75 | | | | | |
|  | R | 3.49 | 3.60 | 10.59 | | | | | |
|  | F value | 0.874 | 0.854 | 7.292 | | | | | |

necessary to determine and apply the appropriate ratio of N, P and K fertilizer to achieve the effect of increasing yield and saving fertilizer.

## Conclusion

The effects of different fertilization formula on blueberry fruit yield and quality were diverse, and its yield and quality may be high or low, but all treatments could improve them by different degrees. This is related to the amount and proportion of different fertilizers. Moreover,

**Table 7. Comprehensive evaluation of the effects of nitrogen, phosphorus and potassium fertilization on the yield and quality of blueberry fruit(2016).**

| Code | Single fruit weight (g) | Yield per plant (kg) | Anthocyanin (mg/g) | Total phenol (mg/g) | Flavonoid (mg/g) | Soluble protein (mg/g) | Soluble solid (%) | Soluble sugar (%) | Titratable acidity (%) | Sugar acid ratio | Comprehensive score Y | ranking |
|---|---|---|---|---|---|---|---|---|---|---|---|---|---|
| F1 | 1.56 | 1.70 | 2.56 | 1.36 | 0.31 | 0.74 | 12.33 | 9.66 | 0.64 | 19.41 | 25.33 | 9 |
| F2 | 1.53 | 2.00 | 3.15 | 1.65 | 0.31 | 0.83 | 13.33 | 10.96 | 0.50 | 27.24 | 30.34 | 1 |
| F3 | 1.44 | 1.32 | 2.40 | 2.09 | 0.35 | 0.83 | 12.67 | 11.60 | 0.54 | 23.58 | 27.37 | 5 |
| F4 | 1.50 | 1.10 | 3.84 | 2.39 | 0.37 | 0.83 | 13.83 | 11.38 | 0.55 | 25.58 | 29.28 | 2 |
| F5 | 1.52 | 1.71 | 3.08 | 2.00 | 0.42 | 0.98 | 13.33 | 13.18 | 0.58 | 23.73 | 28.69 | 3 |
| F6 | 1.60 | 1.53 | 1.55 | 2.35 | 0.38 | 0.96 | 12.00 | 9.74 | 0.66 | 18.26 | 25.40 | 8 |
| F7 | 1.62 | 1.63 | 2.13 | 1.75 | 0.42 | 1.07 | 12.67 | 10.05 | 0.55 | 23.28 | 28.25 | 4 |
| F8 | 1.54 | 1.69 | 2.60 | 2.40 | 0.38 | 1.15 | 12.50 | 12.65 | 0.67 | 18.71 | 26.13 | 6 |
| F9 | 1.48 | 1.12 | 1.91 | 1.74 | 0.38 | 0.59 | 11.83 | 10.32 | 0.51 | 23.44 | 26.06 | 7 |
| CK | 1.49 | 1.10 | 2.05 | 1.17 | 0.26 | 0.62 | 11.13 | 10.18 | 0.60 | 18.67 | 22.76 | 10 |
| Y | K1 | 27.68 | 27.62 | 25.62 | K>P>N | | | | | | | |
|  | K2 | 27.79 | 28.39 | 28.56 | | | | | | | | |
|  | K3 | 26.82 | 26.28 | 28.10 | | | | | | | | |
|  | R | 0.97 | 2.11 | 2.94 | | | | | | | | |
|  | F value | 0.642 | 2.575 | 5.638 | | | | | | | | |

heavy fertilization does not mean that the yield and quality of blueberry fruits must be high. Only a scientific and reasonable fertilization ratio can result in high-yield and good-quality blueberries. Therefore, through comprehensive evaluation and variance analysis, this study concluded that the best combination of N, P and K fertilization formula for blueberry cultivation with high yield, quality and efficiency in this area was N1P2K2 (F2), that is, 100, 25 and 25 g/plant of N, $P_2O_5$ and $K_2O$, respectively.

In conclusion, rational and optimized application of N, P and K fertilizer can achieve good-quality and high-yield of blueberry. In production practice, the optimal fertilization scheme should be determined according to the soil fertility status, referring to the optimal nitrogen, phosphorus and potassium fertilization amount in this study, and based on the principle of increasing N and K, and stabilizing P.

## Author Contributions

**Data curation:** Xinyu Zhang, Shuangshuang Li.

**Formal analysis:** Xinyu Zhang, Xiaoli An, Zejun Song, Yunzheng Zhu, Yi Tan, Xiaolan Guo.

**Funding acquisition:** Delu Wang.

**Resources:** Delu Wang.

**Supervision:** Delu Wang.

**Writing – original draft:** Xinyu Zhang.

**Writing – review & editing:** Delu Wang.

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
