## [Decision Letter · Decision Letter 0]

28 Nov 2022

PONE-D-22-30089Effects of nitrogen, phosphorus and potassium formula fertilization on the yield and quality of blueberry fruitPLOS ONE

Dear Dr. Wang,

Thank you for submitting your manuscript to PLOS ONE. After careful consideration, we feel that it has merit but does not fully meet PLOS ONE’s publication criteria as it currently stands. Therefore, we invite you to submit a revised version of the manuscript that addresses the points raised during the review process.

We look forward to receiving your revised manuscript.

Kind regards,

Sushanta Kumar Naik, PhD

Academic Editor

PLOS ONE

“Author thanks the National Natural Science Foundation of China (NSFC 31260192) for their financial support.”

“Author thanks the National Natural Science Foundation of China (NSFC 31260192) for their financial support.

“Author thanks the National Natural Science Foundation of China (NSFC 31260192) for their financial support.”

6. Please remove your figures from within your manuscript file, leaving only the individual TIFF/EPS image files, uploaded separately. These will be automatically included in the reviewers’ PDF.

Additional Editor Comments:

Give justification that how a one-year trial conducting at a single experimental site is suitable to improve the production of blueberry outside that site.

Several experiments need to be conducted across different agroecosystems due to varying yield-impacting factors to conclude for its recommendation. How is your present investigation justify the fertilizer recommendation across the region?

Re-write the entire manuscript based on proper English usage or consult to a professional.

Reviewers' comments:

Reviewer's Responses to Questions

**Comments to the Author**

1. Is the manuscript technically sound, and do the data support the conclusions?

Reviewer #1: Partly

Reviewer #2: Yes

2. Has the statistical analysis been performed appropriately and rigorously? 

Reviewer #1: Yes

Reviewer #2: I Don't Know

3. Have the authors made all data underlying the findings in their manuscript fully available?

Reviewer #1: Yes

Reviewer #2: Yes

4. Is the manuscript presented in an intelligible fashion and written in standard English?

Reviewer #1: Yes

Reviewer #2: No

5. Review Comments to the Author

Reviewer #1: This is a good attempt to show the importance of major nutrients in production and quality. However, some important aspects are missing in this manuscript which is mentioned below;

• What is the basis of selection of three levels of N, P and K (based upon the recommendation like optimum, sub-optimum or soil test values etc.), as no information was provided on the soil properties.

• Beside the primary nutrients, micro nutrient affects the quality very significantly, but nothing was informed in this aspect.

• The experiment was carried out in 2014, the result presented is based on one year data or else.

• The work was related with the application of N,P, K on yield and quality, but use/ recommendation of these fertilizer only, may create the imbalance of micro nutrient and other soil properties. Which is an important aspect that is missing

• Based upon comprehensive score, it was recommended that F2 is superior but F4 and F9 is second and third, F2 and F4 were very similar. So it is better to compare the benefit cost ratio also before recommendation.

Reviewer #2: In general, the manuscript needs extensive revision for English language and grammar by a professional.

Page 1, title: … and potassium fertilization formula on the yield and berry quality of blueberry Page 2, line 21: …of nitrogen (N), phosphorus (P) and potassium (P) in the field…

From this line and throughout the rest text you must use the above abbreviations for the three nutrients.

Page 2, line 23: blueberry fruits, aiming to optimize the amount of supplied fertilizers.

Page 2, line 28: Even the treatment with the worst results (N2P3K1), …

Page 2, line 30: times higher than the control, respectively.

Page 2, line 33: blueberries was that including N 100 g/plant, P2O5 25 g/plant, K2O 25 g/plant, applied in the form of ammonium sulfate (472 g/plant), superphosphate (41 g/plant) and potassium sulfate (40 g/plant), respectively.

Page 3, line 41: potassium are well-known mineral … development [1], and the application of fertilizers containing these elements can significantly improve the yield…

Page 3, line 43-51: These lines must be written better.

Page 3, line 58: but the yield per unit area and quality are not

Page 4, line 65: “…they often have the problem …potassium fertilizer.” What do you mean?

Page 4, line 68: fertilization affect negatively the yield

Page 4, line 69: and the vegetative growth of blueberries is inhibited, or even damaged [17].

Page 4, line 71: cannot

Page 4, line 72: but also it can improve their fruit…

Page 4, line 76: have pointed out that applying conventional fertilizers without guidance, can lead to salt accumulation in soil and reduce…

Page 4, line 77-78: Delete ‘by applying … habits [8].’

Page 4, line 80: single-nutrient fertilizer

Page 5, line 83: rabbiteye blueberry cv. Brilliant plants, explores

Page 5, line 85: test, aiming to optimize the

Page 5, line 90: 2.1 Experimental site

Page 5, line 91: Please add the geographic coordinates of this site

Page 5, line 92: which is characterized by subtropical

Page 5, line 95: Delete the second ‘soil’

Page 5, line 96-97: Express these values as per cent (%)

Page 5, line 98: 2.2 Plant material

Refer whether these plants were self-rooted or grafted. Were the trees trained with a single trunk or what?

Page 5, line 99: Five -year-old rabbiteye blueberry (Vaccinium ??? cv. Brightwell) plants

Page 5, line 101: 2.3 Experimental design

Refer the plantation distances among trees.

Page 5, line 102: …under field conditions. Fertilizers containing N, P and K doses at three doses for each nutrient, three …

Page 6, line 104: Non fertilized plants were used …

Page 6, line 105: The fertilizers used were…

Page 6, line 106-107: Delete the words ‘content’

Page 6, line 107: …63.2%), all of high chemical purity.

Page 6, line 109: Do you mean ‘fruit production’ or ‘fruit set’?

Page 6, line 110: What do you mean with ‘reducing fertilizer’?

Page 6, line 110-111: The fertilizers were applied into a trench

Page 6, line 112-113: canopy projection, mixed and covered with soil.

Page 6, line 113: What you mean with ‘carried out’?

Page 6, line 114: the experimental design is shown

Page 7, Table 2: There are not all possible N-P-K combinations (eg N3P3K3). How did you select these ones?

Page 7, Table 2: You have not explained what is orthogonal test design. Provide a relative reference.

Page 7, Table 2: I propose the following legend: Table 2. Treatments of N, P and K with their dosages on blueberry plants.

Page 7, Table 2: It would be better to present the doses of nutrients (N, P2O5, K2O) not the form of fertilizers (ammonium sulfate etc)

Page 7, line 126: The Folin-Ciocalteu…method was used to establish …. (mg/g FW)[23].

Page 7, line 128: Replace ‘modified’ with ‘determined’

Page 8, line 133: Refer the type and model of the refractometer device.

Page 8, line 136: …SPSS 18 software programs were used…

Page 8, line 137: ANOVA and least significant difference (LSD) were applied to compare

Page 8, line 138: Delete ‘calculate’

Page 8, line 139: levels was calculated and the principal …evaluation was used.

Page 8, line 141: Results and Discussion

Page 8, line 145: effects were observed in F2 and F6 treatments, which gave 8.67% higher values than ...

Page 8, line 149: best fertilizer combinations affecting the fruit weight and yield…

Page 9, line 152: P>K>N, respectively…

Page 9, line 154: Delete ‘This is the’

Page 9, line 156: The Table 3 repeats data that has been already

Page 9-10, line 156: In the Table 3, column ‘Code’, there is a Chinese character.

Page 10, line 160, 166, 168: sugar: acid

Page 10, line 172: titratable acidity of other

Page 10, line 173: but there is no

Page 11, line 174: effect was that of treatment F9 (17.07% lower than CK)(Fig 2. G)

Page 11, line 176-177: F2 was significantly higher (36.36%) than CK…

Page 11, line 180: is significantly higher (20.59%) than CK…

Page 11, line 187: on the above-mentioned parameters of blueberry fruit is…

Page 11, line 189: ,respectively. According…

Page 12, line 194: Effect of different fertilization formulas on blueberry fruit quality.

Page 12, line 196: Table 4. Orthogonal test on …

Page 13, line 199: yield and berry quality of blueberry.

Page 13, line 200: The yield and berry quality of blueberry were evaluated…

Page 14, line 203-204: Limit to two decimal digits

Page 14, line 212: treatments F2…

Page 14, line 213: were ranked second.

Page 14, line 215, 216, 218: and berry quality

Page 14, line 217: Delete ‘which was also F2’

Page 15, table 5: Titratable acidity

Page 16, line 223: yield and fruit quality of trees. Studies have shown that these nutrients and their…

Page 16, line 226: improve fruit quality

Page 16, line 233-234: it is not clear what you mean. Was the level of soil K in the region low?

Page 16, line 235-236: In accordance, we found that …

Page 16, line 237: titratable acidity in blueberry fruit. There was also significant improvement of the flavonoid…

Page 16, line 238 and 240: sugar: acid

Page 16, line 239: Delete ‘on’

Page 16, line 240: fertilizer on the soluble sugar content, Vc and sugar: acid ratio.

Page 16, line 241: Since K can participate in…

Page 17, line 242: sugar, it increases the sugar…promotes …

Page 17, line 246: Replace ‘extremely’ with ‘highly’

Page 17, line 246: It was shown that the

Page 17, line 248: It was shown that the

Page 17, line 252: and applying more (but not excessive) P fertilizer …

Page 17, line 252-253: Delete ‘but it does not mean adding excessive phosphorus fertilizer.’

Page 17, line 256: low demands for

Page 17, line 259: and berry quality of blueberry, it is necessary to determine and apply the appropriate…

Page 18, line 263: fertilization formula

Page 18, line 264: but all treatments could improve them by …

Page 18, line 265-266: Moreover, heavy fertilization does not mean …

Page 18, line 267: Replace ‘obtain’ with ‘result in’

Page 18, line 271-275: Replace ‘that is, the best … potassium sulfate’ with ‘that is, 100, 25 and 25 g/plant of N, P2O5 and K2O, respectively.’

Page 18, line 277-281: It is not clear what you mean here. Rewrite it better.

Page 19, references: All latin names of plants must be written in italics.

6. PLOS authors have the option to publish the peer review history of their article (what does this mean?). If published, this will include your full peer review and any attached files.

Reviewer #1: No

Reviewer #2: No

---

## [Author Response · Author response to Decision Letter 0]

28 Jan 2023

We have consider all the suggestions and revised the manuscript as per reviewer comments. Response sheet is uploaded in separate file.

---

## [Decision Letter · Decision Letter 1]

13 Feb 2023

PONE-D-22-30089R1Effects of nitrogen, phosphorus and potassium formula fertilization on the yield and berry quality of blueberryPLOS ONE

Dear Dr. Delu Wang,

Thank you for submitting your manuscript to PLOS ONE. After careful consideration, we feel that it has merit but does not fully meet PLOS ONE’s publication criteria as it currently stands. Therefore, we invite you to submit a revised version of the manuscript that addresses the points raised during the review process.

We look forward to receiving your revised manuscript.

Kind regards,

Sushanta Kumar Naik, PhD

Academic Editor

PLOS ONE

Journal Requirements:

Reviewers' comments:

Reviewer's Responses to Questions

**Comments to the Author**

1. If the authors have adequately addressed your comments raised in a previous round of review and you feel that this manuscript is now acceptable for publication, you may indicate that here to bypass the “Comments to the Author” section, enter your conflict of interest statement in the “Confidential to Editor” section, and submit your "Accept" recommendation.

Reviewer #1: (No Response)

Reviewer #2: (No Response)

2. Is the manuscript technically sound, and do the data support the conclusions?

Reviewer #1: (No Response)

Reviewer #2: Partly

3. Has the statistical analysis been performed appropriately and rigorously? 

Reviewer #1: (No Response)

Reviewer #2: Yes

4. Have the authors made all data underlying the findings in their manuscript fully available?

Reviewer #1: (No Response)

Reviewer #2: Yes

5. Is the manuscript presented in an intelligible fashion and written in standard English?

Reviewer #1: (No Response)

Reviewer #2: Yes

6. Review Comments to the Author

Reviewer #1: 1. Soil pH is in acidic range, so significant amount of the applied phosphorus may be fixed in soil, so, whether, it is good to apply P fertilizer directly without any soil amendments.

2. It is still not mentioned in the manuscript that, what is the basis of selection of three levels of N, P and K, How you get the level 1.

3. Selection of recommendation should be based on the B: C ratio or some other important aspects. Which is not mentioned clearly?

4. Presentation of results in figures and tables are complicated, so, if possible convert it in simple form.

Reviewer #2: There are yet some missing data as follows:

Materials and methods

If available, provide the mean nutrient status of tested plants before fertilizer treatments took place.

If available, provide data about the fertilizers applied the year before experimentation.

What were the plantation distances and plantation density (number of trees/ha) of the tested orchard?

What about canopy training system and mean height of the tested trees?

Page 2, line 11: Replace ‘plants’ with ‘species’

Page 2, bottom line: Replace ‘was’ with ‘were’

Discussion, line 11: Delete the word ‘fertilizer’

Conclusion, first line: …formula on blueberry fruit …

Conclusion, bottom line: …of increasing N and K, and stabilizing P.

7. PLOS authors have the option to publish the peer review history of their article (what does this mean?). If published, this will include your full peer review and any attached files.

Reviewer #1: No

Reviewer #2: No

---

## [Author Response · Author response to Decision Letter 1]

16 Feb 2023

Dear Editor: Sushanta Kumar Naik and Dear reviewers:

Thank you very much for your useful comments and professional advice on our manuscript. We wish to give a sincere gratitude to referees for reviewing our paper carefully. These opinions help to improve academic rigor of our article, and we apologize for any inconveniences caused by these errors. We have modified the manuscript accordingly, and the response to the referees’ comments are listed point by point below:

Reviewer #1: 

Comment 1: Soil pH is in acidic range, so significant amount of the applied phosphorus may be fixed in soil, so, whether, it is good to apply P fertilizer directly without any soil amendments.

Reply 1: Thanks for your comment. Although a large amount of phosphorus is fixed in the soil under acidic conditions, it is not the best to apply phosphorus fertilizer directly. Because applying only one kind of fertilizer will aggravate the imbalance of soil nutrients. Moreover, a large number of studies have shown that compared with the single application of nitrogen, phosphorus and potassium, the combined application can significantly improve the fertilizer utilization rate.

Comment 2: It is still not mentioned in the manuscript that, what is the basis of selection of three levels of N, P and K, How you get the level 1.

Reply 2: Thanks for your comment. Select the three factors of N fertilizer, P fertilizer and K fertilizer, and set three levels for each factor according to Guo et al. [22], which has been added to line 3 on page 3 of the manuscript. 

Comment 3: Selection of recommendation should be based on the B: C ratio or some other important aspects. Which is not mentioned clearly?

Reply 3: Thanks for your suggestion. It is calculated that the benefit-cost ratio of treating F2 and F4 is 16.13 and 15.87, respectively, so F2 processing is more recommended, and the relevant content has been added to lines 16-17 on page 6 of the manuscript. 

Comment 4: Presentation of results in figures and tables are complicated, so, if possible convert it in simple form.

Reply 4: Thanks for your suggestion. But I can't find a better way to show the results. If possible, I hope you can give me some advice. 

Reviewer #2: There are yet some missing data as follows: Materials and methods

Comment 1: If available, provide the mean nutrient status of tested plants before fertilizer treatments took place.If available, provide data about the fertilizers applied the year before experimentation.What were the plantation distances and plantation density (number of trees/ha) of the tested orchard?What about canopy training system and mean height of the tested trees?

Reply 1: Thanks for your suggestion. We regret that we did not provide enough complete material information in the materials and methods section. First of all, the plantation distances and plantation density (number of trees/ha) of the tested orchard are 1.5m and 4444 trees/ha respectively, which have been added to lines 4-5 on page 3 of the manuscript. Secondly, as a response to the reviewers, we regret the lack of detailed data on the mean nutrient status of tested plants before fertilizer treatments took place, the data of the fertilizers applied the year before experimentation, the canopy training system and mean height of the tested trees. This is a mistake in our research. We will be more cautious in future research.

Comment 2: Page 2, line 11: Replace ‘plants’ with ‘species’.

Reply 2: Thanks for your suggestion. The relevant content has been revised in line 10 on page 2 of the manuscript.

Comment 3: Page 2, bottom line: Replace ‘was’ with ‘were’.

Reply 3: Thanks for your suggestion. The last line on page 2 of the manuscript has been revised.

Comment 4: Discussion, line 11: Delete the word ‘fertilizer’.

Reply 4: Thanks for your suggestion. Relevant contents are modified in line 11 of the discussion part of the manuscript.

Comment 5: Conclusion, first line: …formula on blueberry fruit …

Reply 5: Thanks for your suggestion. Relevant contents are revised in the first line of the conclusion of the manuscript.

Comment 6: Conclusion, bottom line: …of increasing N and K, and stabilizing P.

Reply 6: Thanks for your suggestion. The last line of the conclusion of the manuscript has been revised.

---

## [Decision Letter · Decision Letter 2]

3 Mar 2023

Effects of nitrogen, phosphorus and potassium formula fertilization on the yield and berry quality of blueberry

PONE-D-22-30089R2

Dear Dr. Wang,

We’re pleased to inform you that your manuscript has been judged scientifically suitable for publication and will be formally accepted for publication once it meets all outstanding technical requirements.

Kind regards,

Sushanta Kumar Naik, PhD

Academic Editor

PLOS ONE

Additional Editor Comments (optional):

Reviewers' comments:

Reviewer's Responses to Questions

**Comments to the Author**

1. If the authors have adequately addressed your comments raised in a previous round of review and you feel that this manuscript is now acceptable for publication, you may indicate that here to bypass the “Comments to the Author” section, enter your conflict of interest statement in the “Confidential to Editor” section, and submit your "Accept" recommendation.

Reviewer #1: (No Response)

2. Is the manuscript technically sound, and do the data support the conclusions?

Reviewer #1: (No Response)

3. Has the statistical analysis been performed appropriately and rigorously? 

Reviewer #1: (No Response)

4. Have the authors made all data underlying the findings in their manuscript fully available?

Reviewer #1: (No Response)

5. Is the manuscript presented in an intelligible fashion and written in standard English?

Reviewer #1: (No Response)

6. Review Comments to the Author

Reviewer #1: (No Response)

7. PLOS authors have the option to publish the peer review history of their article (what does this mean?). If published, this will include your full peer review and any attached files.

Reviewer #1: No

---

## [Editor Report · Acceptance letter]

7 Mar 2023

PONE-D-22-30089R2 

Effects of nitrogen, phosphorus and potassium formula fertilization on the yield and berry quality of blueberry 

Dear Dr. Wang:

I'm pleased to inform you that your manuscript has been deemed suitable for publication in PLOS ONE. Congratulations! Your manuscript is now with our production department. 

Kind regards, 

on behalf of

Dr. Sushanta Kumar Naik 

Academic Editor

PLOS ONE